# Dynamic Learning Framework for Smooth-Aided Machine-Learning-Based Backbone Traffic Forecasts

**DOI:** 10.3390/s22093592

**Published:** 2022-05-09

**Authors:** Mohamed Khalafalla Hassan, Sharifah Hafizah Syed Ariffin, N. Effiyana Ghazali, Mutaz Hamad, Mosab Hamdan, Monia Hamdi, Habib Hamam, Suleman Khan

**Affiliations:** 1School of Electrical Engineering, University Technology Malaysia, Skudai, Johor 81310, Malaysia; sharifah@fke.utm.my (S.H.S.A.); nurzal@utm.my (N.E.G.); 2School of Telecommunication Engineering, Future University, Khartoum 10553, Sudan; hhkmutaz2@graduate.utm.my; 3Department of Computer Science, University of São Paulo, São Paulo 05508-090, Brazil; mosab.hamdan@ieee.org; 4Department of Information Technology, College of Computer and Information Sciences, Princess Nourah Bint Abdulrahman University, P.O. Box 84428, Riyadh 11671, Saudi Arabia; mshamdi@pnu.edu.sa; 5Faculty of Engineering, Uni de Moncton, Moncton, NB E1A3E9, Canada; habib.hamam@umoncton.ca; 6Department of Electrical and Electronic Eng. Science, School of Electrical Engineering, University of Johannesburg, Johannesburg 2006, South Africa; 7International Institute of Technology and Management, Commune d’Akanda, Libreville BP 1989, Gabon; 8School of Psychology and Computer Science, University of Central Lancashire, Preston PR1 2HE, UK; skhan92@uclan.ac.uk

**Keywords:** traffic forecast, slice, local smoothing, LSTM, dynamic learning

## Abstract

Recently, there has been an increasing need for new applications and services such as big data, blockchains, vehicle-to-everything (V2X), the Internet of things, 5G, and beyond. Therefore, to maintain quality of service (QoS), accurate network resource planning and forecasting are essential steps for resource allocation. This study proposes a reliable hybrid dynamic bandwidth slice forecasting framework that combines the long short-term memory (LSTM) neural network and local smoothing methods to improve the network forecasting model. Moreover, the proposed framework can dynamically react to all the changes occurring in the data series. Backbone traffic was used to validate the proposed method. As a result, the forecasting accuracy improved significantly with the proposed framework and with minimal data loss from the smoothing process. The results showed that the hybrid moving average LSTM (MLSTM) achieved the most remarkable improvement in the training and testing forecasts, with 28% and 24% for long-term evolution (LTE) time series and with 35% and 32% for the multiprotocol label switching (MPLS) time series, respectively, while robust locally weighted scatter plot smoothing and LSTM (RLWLSTM) achieved the most significant improvement for upstream traffic with 45%; moreover, the dynamic learning framework achieved improvement percentages that can reach up to 100%.

## 1. Introduction

Next-generation networks have been designed to offer reliable service with ultra-low latency, massive-scale connectivity, high security, extreme data rates, optimized energy, and better quality of service (QoS) [1,2,3]. Despite these features, the technology (infrastructure and logic) used in these networks must display an intelligence for coping with the dynamic QoS demand [4,5,6,7,8,9] and react autonomously to different dynamic and self-organizing situations. Additionally, network management is complicated due to the coupling between various service layers where congestions can arise and spread vertically as well as horizontally. Furthermore, the congestions arising due to poor management can affect the QoS and service-level agreement (SLA). Therefore, proactive approaches for managing bandwidth and network resources are highly needed. The legacy static network resource allocation indicated that a bandwidth reservation can guarantee a particular QoS. However, a dynamic network resource allocation effectively resolves this problem [10,11,12,13]. It relies on the forecasting network resources’ demands and acts accordingly to enable a timely and dynamic response. Thus, the accuracy of predictive approaches was regarded as a vital factor and essential in various applications of the predictive frameworks. Reliable artificial intelligence (AI) and machine learning (ML) techniques are crucial and widely used in different applications, such as network traffic forecasts [4,5,6,7,8,9,14], the Internet of things (IoT) [10], and wireless communications [11,15]. The data characteristics indicated that the traffic used in real-time applications in current and future networks exhibited variable, nonlinear, and unstructured data formats with slowly decaying autocorrelations between different samples. These features showed that the traffic can exhibit long-range dependence (LRD) [12,16]. To ensure a proper control strategy, the short-step forecasts, such as the one-step forecast models used for LRD traffic, could not respond accurately to the dynamic bandwidth allocation, especially in the higher latency links [1]. Hence, a long-term traffic forecast was required for implementing a flexible strategy to control the networks [1].

Many ML-based studies have focused on the multiple-step bandwidth forecasting process. In general, two different approaches were used for designing bandwidth forecast algorithms, where the first algorithms were based on the supervised ML models, the other relies on statistical models. The first algorithm described various traffic forecasting models based on a supervised ML process, specifically the artificial neural networks (ANNs). On the other hand, the second algorithm used statistical models that are based on the generalized autoregressive integrated moving average (ARIMA) model [1,2]. The major difference noted between the ANN and ARIMA models was that the ARIMA model required the imposition of a stationary property. It also did not accurately forecast while handling LRD [17,18]. The LSTM process is a very effective ML technique used for time series forecasts. This process was applied in multiple-step predictions under different scenarios. The main advantage noted after applying the LSTM-RNN process was that it could quickly learn the temporal dependencies on the input data. At the same time, it was not necessary to specify a fixed set of lagged inputs [17,18,19]. LSTM could resolve the long-term dependency issue as it memorized the information for more extended periods, unlike some other linear time series forecast algorithms (such as ARIMA and its various extensions) that were affected by the unnecessary fluctuations occurring in the series [19,20,21,22]. This is then projected to all forecasted results. Owing to the ability of the LSTM technique to forget or remember information based on its activation function, these techniques can re-evaluate their weights based on their correlation with the remaining time series. This ability of LSTM makes it versatile and adaptable when dealing with errors, noise, and sample gaps. However, as the fluctuations and noise increase in a time series, it is more difficult for the forecasting technique to provide accurate performance. Therefore, data smoothing and filtering must be conducted before any forecasting. This preprocessing method could handle significant fluctuations and outliers by adjusting the built-in sliding window. Motivated by these, we consider the combination of LSTM and smoothing for the multistep-ahead forecasting of backbone network traffic forecasting.

Moreover, to address model reliability and validity, the concept changes detection mechanism must be incorporated and addressed due to the rapid data characteristics and distribution changes. In this work, a real dataset was collected and analyzed. The dataset was collected from a premier internet service provider backbone network. The major contributions of the study can be summarized as follows:Investigation of the hybrid multistep-ahead forecast framework after combining LSTM and the local smoothing techniques for the network traffic forecast;A change detection framework is proposed. This framework was used to determine when to build new hybrid forecast model;Finally, the effectiveness of the model was furtherly analyzed and compared with the relevant study.

The remaining study is organized in the following manner: Section 2 discusses the related works. Section 3 provides a detailed description of the smoothening-aided LSTM model for bandwidth slice forecasts. Then, Section 4 discusses the performance of the proposed model. It also shows the forecasting accuracy, smoothing analysis, and statistical validation of the results. Lastly, the conclusion is presented in Section 5.

## 2. Related Work

Several studies have analyzed the effectiveness and superiority of the LSTM process for bandwidth forecasting [21,22,23,24,25,26,27,28,29,30]. For instance, in [28] the researchers investigated the performance of different ML techniques and assessed the forecast performances of their video over the internet. They studied neural networks (NNs), support vector machines (SVMs), and decision trees (DTs). They concluded that modeling based on the time series data was better for generating promising results. Additionally, it was seen that the ANN model showed a better performance than the other ML techniques. In [24], the researchers used a hybrid neural network-wavelet model to analyze network traffic. They used the wavelets for decomposing the input data into details and approximations, while the NN was optimized with the help of a genetic algorithm. They noted that their proposed model could significantly improve the forecast accuracy of the process. Though wavelet processing helps in eliminating the unnecessary data, it can lead to some unintentional issues via the traffic load forecast based on LSTM and deep NNs (DNN). The simulation results showed that the forecast-based scalability mechanism performed better than the threshold-based one. In [23], the researchers proposed a new mechanism for scaling the access management functions (AMFs) in the 5G virtualized environment. This mechanism was based on forecasting the mobile traffic using the LSTM NNs to estimate the user attach request rate, which helped predict the accurate number of AMF examples required to process the upcoming user traffic. Since it is a proactive technique, the proposed model helps avoid deployment latency while scaling the resources. The simulation results further indicated that the LSTM-based model was more efficient than the threshold-based model. The proposed technique used LSTM on the request rate data without preprocessing, which eventually may decrease the forecast accuracy. In [21], the researchers compared the performance of the LSTM networks used for 4G traffic forecasts, seasonal ARIMA (SARIMA), and the support vector regression (SVR). For this purpose, they collected the data for 122 days, for which the data points were divided between the training and testing datasets. They noted that the LSTM model showed better performance than the SARIMA and SVR networks. In [26], the researchers developed a deep traffic predictor (DeepTP) model to forecast long-period cellular network traffic. They noted that their model showed better performance (12.3%) than the other traffic forecasting models used in the study. Furthermore, a feature-based forecasting framework that used tier 1 internet service provider (ISP) network traffic was discussed. LSTM was used as the core forecast technique. The results obtained were significant and forecasted the traffic at very small time scales (<30 s). In [22,29], the researchers discussed and proposed a hybrid empirical mode decomposition (EMD) and LSTM forecast technique. EMD decomposes the available bandwidth dataset into smoothened interstice mode functions (IMFs). After that, they applied LSTM to forecast the traffic. They noted that their hybrid model showed a better root mean square error (RMSE) value. In different studies [27,30], the authors used LSTM for forecasting the vehicular ad hoc network (VANET). They determined the forecasting accuracy with the help of the RMSE and mean absolute percentage error (MAPE); the results proved the effectiveness of the proposed mechanism. The authors of [31] proposed a smooth-aided SVM-based model for video traffic forecasting; the obtained results were promising where local smoothing techniques were incorporated ahead to the SVM to normalize the fluctuations in the input traffic. The smoothed support vector machine (SSVM) has an improvement percentage of 32.35% for a one-step-ahead forecast. In the most recent study [32], the authors proposed a hybrid LSTM and convolution neural network framework for a wireless network. The proposed solution was compared with state-of-the-art techniques, and the effectiveness and superiority of the hybrid architecture were highlighted. Table 1 summarize the most notable related work.

Though the earlier studies presented many positive results, we noted that the accuracy of multistep-ahead forecasting in autonomous network management was very challenging. Owing to the noise inconsistency and bursts in the network traffic, small fluctuations occurred in the traffic data that could degrade the forecast accuracy of the model [18]. Very few studies reported in the literature considered noise preprocessing, while no study presented a dynamic framework for concept changes. Previously adopted noise preprocessing methods in previous studies, such as wavelets and EMD, are less flexible than window-based noise processing [31]. The only notable study used windows-based noise processing, such as Gaussian smoothing, moving average, and Savitzky–Golay filters, and used SVMs as an ML technique. At the same time, it was already proven in [28] that NNs and LSTM outperform SVMs in forecasting accuracy. Moreover, the mentioned study was carried out in very limited scenario (one-step forecasts).

Due to the evolved dynamic nature of the network properties, the frameworks must detect and adapt to all changes taking place in the statistical properties of the big data traffic. The changes noted in the traffic profiles, such as a sudden surge in the traffic, took place due to the change in the users, application behavioral variations taking place in the traffic demands, and because of the emergence of novel technologies, applications, or even a global pandemic, such as that of COVID-19 [33,34]. As a result, the number of home users or eMBB traffic increases significantly compared with the corporate traffic [33], which witnessed a significant decrease owing to lockdowns and widespread adoption of work-from-home culture in the business operational model. In this study, considering the promising finding of using window-based techniques as a preprocessing method to handle all the significant fluctuations and outliers by adjusting the built-in sliding window, we extended these results to further these studies and explore the effects of hybrid local smoothing processes and the LSTM-NN technique [35].

## 3. Methods

To resolve the challenges related to resource management noted in next-generation network backbones, i.e., a beyond 5G (B5G) network environment, we propose a hybrid ML model. This model combines the LSTM and smoothing processes and uses them for the core network bandwidth slices. The forecasting model is called the smoothed LSTM. Figure 1 shows the proposed overall conceptual framework. The model is motivated by the promising results presented earlier [31]. The proposed ML technique is modeled as a time series batch learning process. The researchers extended this algorithm by preprocessing the dataset.

As depicted in Figure 1, the Anderson–Darling test was employed as a change detection method to dynamically manage the dynamic selection of the hybrid algorithm based on the changes in the underlying statistical properties. Then, to avoid eroding the periodic patterns and trends in the series, the system studied the local and global trends separately to detect and eliminate long-term or short-term noise. The preprocessing focuses on the local variations. It applied local smoothing techniques to eliminate the fluctuations and unnecessary noise in the data, which can negatively affect the model’s prediction accuracy, especially in the case of the nonlinear and nonstationary time series. The local preprocessing techniques show a higher dynamic reaction to the noise level and short-term variations than the other wavelet- and Hilbert–Huang transform (HHT)-based processes. A similar approach was used earlier [31], where researchers studied the superior nonlinear approximation ability of an SVM combined with the “classical” local smoothing processes, such as Gaussian smoothing, moving average, and Savitzky–Golay filters. The study results indicated that their proposed model performed better than the state-of-the-art model, viz., logistic regression. We determined the effectiveness of their proposed model by using the real and available network traffic datasets. After the local smoothing preprocessing takes place and the provided y arrives as an input, a forecast ${\hat{y}}_{t}$ is produced using the current LSTM model δ, after which a loss function $f\left({\hat{y}}_{t},y_{t}\right)$ is used to update the model. Finally, a statistical test was conducted by the Diebold–Mariano test to validate the obtained results.

### 3.1. Dataset

The dataset was collected from a premier internet service provider in Africa. We examined different bandwidth utilization time series; the collected data represent LTE, MPLS, and the upstream tier 1 carrier traffic’s aggregated backbone traffic. Three hundred and fifty time steps’ sample data were collected. Each time step represents 28.8 min, and 350 time steps represent one week. This was attributed to the limitations of the data collection tool. The values were interpolated and used for developing a time series model. The findings will benefit the real-world core and backbone networks in such a way as to achieve efficient network resource planning.

For this work, the computer specifications that were used to process and execute the proposed framework were Core i5 1.8 GHz with 16 GB of RAM. Figure 2 shows the backbone topology where the dataset was collected.

Table 2 shows the description for each bandwidth slice.

Three different traffic profiles were used to explore different traffic characteristics. Slice 1 represents the aggregated backbone traffic for 4G-LTE measured at the SGI interface between the packet data network (PDN-GW) and the core routers in the evolved packet core (EPC). The EPC is responsible for the establishment, management, and authentication of users’ sessions. The core routers are linked to the MPLS backbone network and the tier 1 upstream providers through the upstream routers. Slice 2 is the aggregated backbone traffic for corporate data centers; it was gathered from corporate users’ virtual routing function (VRF) instances at the MPLE backbone routers. Finally, slice 3 represents aggregated traffic at upstream router (A).

It is evident from Figure 3 that all bandwidth slices exhibit significant seasonal patterns with daily peaks. Nevertheless, the data also show a stochastic pattern with continuous irregular fluctuations between successive points. On the other hand, no long-term trend appeared to exist. Some slices exhibit a weekly pattern, such as in the MPLS slice since it is more associated with corporate users where corporate business is active mainly during weekdays rather than during weekends. Table 3 shows the summarized descriptive statistics, Figure 3 shows the sample time series dataset and Figure 4 shows the dataset histograms.

From Table 3, the most notable statistical property is that the LTE and MPLS follow the Johnson SB statistical distribution, while the upstream traffic follows the Gen. extreme value distribution. Equations (1)–(4) show the portability density functions (PDFs) for each distribution function associated with every bandwidth slice, respectively:
(1)$$f\left(x\right)=\frac{-0.829}{0.589\sqrt{2\pi }z\left(1-z\right)}exp\left(-\frac{1}{2}{\left(0.589-0.829ln\left(\frac{z}{1-z}\right)\right)}^{2}\right)$$ where $z\equiv \frac{x-3.885\times 10^{8}}{0.589}$.
(2)$$f\left(x\right)=\frac{0.854}{0.398\sqrt{2\pi }z\left(1-z\right)}exp\;\left(-\frac{1}{2}{\left(0.3980.854ln\left(\frac{z}{1-z}\right)\right)}^{2}\right)$$ where $z\equiv \frac{x-1.637\times 10^{8}}{0.398}$.
(3)$$f\left(x\right)=\frac{1}{1.08\times 10^{9}}t{\left(x\right)}^{0.48}exp^{-t\left(x\right)}$$ where $t\left(x\right)={\left(1-0.52\left(\frac{x-4.88\times 10^{9}}{1.08\times 10^{9}}\right)\right)}^{\frac{-1}{-0.52}}$


Since ξ≠0
(4)$$\frac{3.13}{5.92\times 10^{7}\Gamma \left(\frac{1}{3.13}\right)}e^{-{\left(\left|\frac{x-6.35)}{2.96\times 10^{7}}\right|\right)}^{3.13}}$$
(5)$$f\left(x\right)=\frac{1.53}{1.39\times 10^{8}\sqrt{2\pi }z\left(1-z\right)}exp\;\left(-\frac{1}{2}{\left(-0.072+1.5ln\left(\frac{z}{1-z}\right)\right)}^{2}\right)$$ where $z\equiv \frac{x-1.11\times 10^{7}}{1.3966\times 10^{8}}$.

### 3.2. Local Smoothing Techniques

As discussed, noise in the time series forecast can significantly and negatively affect the forecasts in the n steps ahead. Hence, this issue must be handled carefully. Minimizing the effects of low- and high-frequency noise can help accurately forecast the short- or long-term-scale data. Some earlier studies discussed the importance of noise removal or data processing [7,18,22,24,29]. In the subsequent section, we discuss the different local smoothing methods applied in this study.

#### 3.2.1. Local Regression

LOWESS [36] is a first-degree polynomial model with weighted linear least squares, while LOESS is a second-degree polynomial model based on the basic fitting model, which employs localized data subsets to construct a curve that approximates the primary data, with weights derived using Equation (6). The LOWESS model evaluates the fit at $x_{i}$ for deriving the fitted values, $\left({\hat{y}}_{i}\right)$, and residuals, ${\hat{\varepsilon }}_{i}=\;{\hat{y}}_{i}\;-\;y_{i}$, at every observation ($x_{i}$, $y_{i}$). The additional robustness weight $w_{i}$, was calculated and subjected to the magnitude of ${\hat{\varepsilon }}_{i}$. Accordingly, a new weight $w_{i}\left(x_{i}\right)$, was assigned to each observation, where $w_{i}$ is defined as shown in Equation (7) [34]:
(6)$$w_{i}\left(x\right)=\left(\frac{\Delta_{i\;}\left(x\right)}{\Delta_{q}\left(x\right)}\right)$$
(7)$$w_{i}=\left\{\begin{array}{c} {\left(1-{\left(\frac{{\hat{\varepsilon }}_{i}}{6MAD}\right)}^{2}\right)}^{2},\left|{\hat{\varepsilon }}_{i}\right|<6MAD\\ 0\;,\;\left|{\hat{\varepsilon }}_{i}\right|\geq 6MAD \end{array}\right\}$$ where $MAD=Median\;\left(\left|{\hat{\varepsilon }}_{i}\right|\right)$.


Two different versions of the above techniques were used, i.e., “RLOWESS” and “RLOESS”. In these forms, the researchers assigned lower weights to the outliers in the regression. Moreover, zero weights were assigned to the new values outside the six mean absolute deviations.

#### 3.2.2. Moving Average

Moving average (MA) [13,35,36,37] is regarded as a real-time filter that eliminates the high frequency from the data. It is generally used for trend forecasting. The estimated coefficients were equal to the reciprocal of the span or bandwidth. MA is also called “exponential smoothing”. Here, the researchers define $C_{i}$ as the throughput at time *i*. Consider $c=\left\{C_{i}\right\},i=1\ldots p$ as the time series, where *p* was the length of the time series. Hence, the MA of period *q* at time l was calculated using Equation (8) [35,36,37,38,39]:
(8)$$m_{l}^{q}=\;\frac{1}{q}\overset{q}{\underset{i=1}{\sum }}c_{l-i+1}$$

#### 3.2.3. Savitzky–Golay Smoothing Filter

The Savitzky–Golay (SG) smoothing filter [40] is a low-pass filter that is characterized by two parameters that are indicated as *K* and *M*. The SG filter is defined as the weighted MA value, i.e., a finite impulse response (FIR) filter. The researchers calculated the filter coefficients using the unweighted linear least squares regression and polynomial model of a particular degree (default of 2). Furthermore, the time series to be determined is described as *x*(*n*), while the observed time series was estimated as *y* (*n*) = *x* (*n*) + *w* (*n*). Here, *w* (*n*) is regarded as the additive white Gaussian noise, wherein the final output is derived using Equation (9):
(9)$$\hat{x}\left(n\right)=\sum_{K=-M}^{M}h\left(k\right)y\left(n-k\right)\;I$$

It is noted that a high-degree polynomial helps in achieving a higher smoothing level without attenuating any data features. It is worth mentioning that LOESS is used for seasonal decomposition. However, we focused on using LOESS and other local regression techniques for smoothing in this study since decomposition may aggressively remove some important dataset features. Let us understand how to choose the bandwidth *q*. Bandwidth plays a vital role in the general local regression fit, while the simplest approach involves selecting *q* as a constant for all $x_{i}$. However, a large variance is observed if the selected bandwidth is minimal. This was attributed to insufficient data falling in the smoothing window and generating a noisy fit. However, not all data will be fitted in the specified window if *q* is very large. As a result, it is challenging to select an optimum *q* value to avoid unnecessary data loss from the original time series. Hence, we proposed a solution, described in Algorithm 1, that finds the minimum *q* value that causes minimal data loss reflected in the minimum mean square error (MSE).
**Algorithm 1** Loss aware smoothing ***Input:*** y:Bandwidth Slice*, Z: Series length, q: smoothing window* *size*
 ***Output:***
 *MSE,*$\;\hat{y}\;:\;$
*locally fitted value using local smoothing technique*
 ***Process:***
 **1***- For n = q to Z − q do*
 **2***- Initialize K [];*
 **3***- for j = n − q to n + q Do*
 **4***-*
$\hat{y}\;\leftarrow \;$*smooth (y*$\left(j\right))\;with\;minimum\;MSE$
 **5***- Assign (*$\hat{y}\left(j\right))\;into\;K$
*[]*
 **6***- Return*
$\hat{y}$


### 3.3. Anderson–Darling

The Anderson–Darling test is [41] a nonparametric test that shows a superior performance while detecting departures from normality [41]. A K-sample is a type of Anderson–Darling test used to detect if multiple observations are generated from the same statistical distribution.
(10)$$AD=-n-\frac{1}{2}\overset{n}{\underset{i\;=\;1}{\sum }}\left(2i-1\right)\left(\ln\left(x_{i}\right)+\ln\left(1-\left(x_{n+1-i}\right)\right)\right)$$ where $\{x_{i\;}$ < … < $\;x_{n}\}$ is the ordered input sample of size *n* (ranging from the smallest to the largest element). The hypothesis states that the $\{x_{i\;}$ < … < $\;x_{n}\}$ that arises from the same distribution is rejected if the AD in Equation (10) *i* larger than the critical values of $AD_{\alpha }$ at the given α.

### 3.4. LSTM

LSTM [32] is a recurrent neural network that forgets and propagates information for a recurrent training period. This can improve the forecast performance. Due to its ability to correlate current and earlier information, the LSTM technique effectively forecasts time series [42]. The cell represents the basic unit of LSTM. Assume *t* as the sequence vector, where *t* = 1, 2, … *T* denotes the sample index, while *T* defines the total time series samples present in a sequence. At every index *t*, the input sample, *xt*; past cell state, *at* − 1; and past hidden state, *ht* − 1; were considered by LSTM. All temporal relationships in LSTM can be derived using the equations below [32,42]:
(11)$$\Gamma f\;t\;=\;\sigma \left(Wfhht-1\;+\;Wfxxt\;+\;bf\right)\;$$
(12)$$\Gamma it\;=\;\sigma \left(Wihht-1\;+\;Wixxt\;+\;bi\right)\;$$
(13)$$\Gamma gt\;=\;\rho \left(Wghht-1\;+\;Wgxxt\;+\;bg\right)$$
(14)$$\Gamma o\;t\;=\;\sigma \left(Wohht-1\;+\;Woxxt\;+\;bo\right)$$
(15)at = Γf t ⨀ at−1 + Γit ⨀ Γg t
(16)$$ht\;=\;\Gamma ot\;⨀\;\rho \left(at\right)$$


Here, *Wfh*, *Wfx*, *Wih*, *Wih*, *Wgh*, *Wgh*, *Woh*, and *Woh* represent the weight matrices, while *bf*, *bi*, *bg*, and *bo* represent the bias vectors that corresponded to the respective resultant vectors for Γ*f t*, Γ*it*, Γ*g t*, and Γ*o t*. Additionally, the forget gate, input gate, input node, and output gate are represented by using the subscript notations of *f*, *i*, *g*, and *o*, respectively. The symbol “⨀” is an elsewise product. In Equations (11)–(16), the researchers represented the weight matrices by *T* × *T*, with a vector size of *T* × 1. The cell state emulated LSTM. The output of the hidden state was considered as a virtual output of the cell state. The sigmoid and rectified linear unit (ReLU) were used as the activation functions in this study; they were represented by σ(*z*) = 1·1 + *e* − *z*, which yields an output in the range of (0, 1) for any input. The activation function can be used across all the LSTM gates, wherein the output gates decide if the data should be propagated (values near 1 or 0). The LSTM training process includes gradient computation that eliminates the gradient problem if all the gradients are reduced to zero [36]. ReLU activation can handle this issue, where gradients are calculated faster. However, they are not easily eliminated [32]. The function of a forget gate is to choose what information to retain and what information to remove from *ht* − 1 and *xt*. This output results in the vector Γ *f t* (11), which contains values ranging between (0 and 1) that help in eliminating the irrelevant values from the cell state. Then, by applying the sigmoid activation, the new information yields indices by the input gate that further yield the vector Γ*it* (12). The output from the ReLU activation encourages the inclusion of new values in the vector Γ*gt* (13). The result of the element-wise product of Γ*it* and Γ*gt* that contains new values is added to Γ*ft*⨀*at* − 1. This provides the updated cell state *at* (15). After this, the filtered value from the updated cell state *at* is passed as the new hidden state *ht*. The values that are passed to the new hidden state *ht* are determined after passing the updated cell state *at* through the ReLU activation. This eventually yields ρ(*at*). Then, we determine the location of all updated cell state vectors which maintain the filtered values by the sigmoid activation (14), resulting in the vector Γ*ot*. Finally, *ht* is seen to be the final hidden state that can be calculated using Equation (16). Algorithm 2 presents the LSTM training process. The training process combines three repetitive processes, i.e., forward propagation, backward propagation, and model updates. The process continues further to minimize the training error. Then, the forward propagation forwards the training sample, X (where X ϵ *y*) and batch size B, with a learning rate of α. The output is then backwards propagated using g, where g ϵ y. After that, the error, E, and learning rate, α, are updated accordingly.
**Algorithm 2** Training Process**Input**:
 y*: bandwidth slice, p: Epochs, B: Batch size, X: training, g: testing,*
α: *learning rate,*
$\hat{\;\vartheta }:$
*Initial Model*
 **Output**:
 ϑ*: LSTM Model, E: Forecast Error, P: parameters*
 **Process:**
 **01**: begin
 **02**: **for** I ← to p
 **03**: ϑ ← *forward propagate (*$\hat{\;\vartheta },$
*X, y,*
α,B) 
 **04: E**←
*Backward Propagation (*ϑ*,g)*
 **05*****: P***$\leftarrow Update\left[\vartheta ,E,\alpha \right]$

 **06: End for**

Similar to the approach used in [23], hyperparameter selection was conducted through a grid search, as depicted in Table 4. This is due to its reliability and simplicity [42]. Other options for hyperparameter selection include random search, Bayesian optimization, particle swarm optimization (PSO), and genetic algorithm (GA) [42,43,44].

### 3.5. Dynamic Learning Framework

Due to the evolved dynamic nature of the existing network properties, we proposed a dynamic framework to detect and adapt to any changes in the data patterns of the data traffic. Concept change is popularly used in statistics and data stream analysis. In this study, Algorithm 3 was presented, wherein the framework consisted of S, local smoothing algorithms, and where $\varphi_{i}$ denotes the hybrid smoothed algorithms formed after combining the local smoothing algorithms and trained LSTM neural network. The input of change detection consists of a bandwidth slice y which is allocated based on the window size $W_{j}.\;t_{i}$ is the time step of the bandwidth slice at index i, where $t_{i}\epsilon \;y$. During the initial stages, the reference δ represents the final selected hybrid algorithm and is initialized at the window, $W_{j}$. The current window slides onto the data series and captures the next batch of data series. After detecting any change, the change detector raises the alarm. However, if no change occurs, the primary window $W_{j}$ slides step-by-step until any change is detected. Here, change refers to a change in the statistical distribution between $W_{j}$ and $W_{j+1}$, defined by the Anderson–Darling test. In general, change detectors are used as a part of online classifiers to guarantee a quick response to sudden changes. If some changes are detected, then it is believed that the existing forecasting algorithms cannot accurately forecast when using the new data as the input. Hence, a new hybrid forecast algorithm must be trained and put in place for the generation of a novel forecasting model.

A smoothed bandwidth slice $\hat{y}$ is used in the next window $W_{j+1}$ by utilizing the local smoothing algorithms in *S*. Then, a new LSTM hybrid model $\;\varphi_{i}$ using Algorithm 2 is developed. After that, the error function *E* is calculated as a testing loss function; a new list of $\varphi_{i}\left[\;\right]$ is sorted with a minimum error function. Then, a statistically significant test is performed using the Diebold–Mariano test for verifying if the new $\varphi_{i\;}$ is statistically different from the other hybrid algorithms in the list. However, if the new hybrid algorithm is better, the old model $\delta_{i-1}$ is replaced with the new one. The pseudocode of the change detection is described in Algorithm 3. Figure 5 depicts the block diagram for the proposed framework.
**Algorithm 3** Dynamic Learning**Input**: 
y*: bandwidth slice, S: list of local smoothing algorithms,*
$\varphi_{i}$
*: hybrid smoothed LSTM algorithm, k: list of hybrid Smoothed LSTM algorithms (6 in this case),* 
**Output**: 
δ*: statistically significant smoothed LSTM algorithm, E: Forecast Error* 
**Process** 
**01**: begin 
**02**: δ←∅; 
**03**: *for all time steps*$\;t_{i}\epsilon \;y\;$
**do** 
**04:**
$W_{j}\leftarrow W_{j}\;\cup \left\{t_{i}\right\}$; 
**05**: **if**
*Change is detected = true*
**then**//using Anderson–Darling 
**06**: *Stop forecasting at*
$W_{j}$ 
**07**: $\hat{y}\leftarrow \;$
*smooth*
y 
*in*
$W_{j+1}$
*using algorithms in S* 
*//algorithm 3* 
**08***: for*
$\varphi_{i}$
*ink* 
**09***:*
$\varphi_{i}\left[\;\right]\leftarrow$
*Build new hybrid LSTM models (*$\hat{y},\vartheta )$ 
*//algorithm 2* 
**10**: *E*
←
*Calculate Forecast error of*
$t_{i}\;in\;W_{j+2}$
*using*
$\varphi_{i}$ 
**11**: *k*$\leftarrow k\cup \;\varphi_{i}\;sorted\;with\;Min\left(E\right)$ 
**12**: δ← 
*Find in k the significant*
$\varphi_{i\;}$
*with Min(E)* 
**13***:*
**If**
δ
*is significantly better than*
$\delta_{i-1}$
*//(old-Existed* *forecast algorithm)*
**then** 
**14**: *replace*$\;\;\delta_{i-1}$
*by*
δ 
**else if** 
**15**: *keep*
δ 
**16**: *endif* 
**17***: endif* 
**18***: Loop*

## 4. Results and Discussion

In this study, the stationarity of time series was confirmed using the augmented Dicky–Fuller (ADF) test as the nonstationary models can yield misleading results, as observed in earlier studies [24,27]; although, LSTM can be used to model a nonstationary time series. Moreover, we normalized the time series variances using the Box–Cox power transformation. Figure 6 presents the bandwidth utilization using the MA smoothing technique, whereas Figure 6a depicts the MPLS bandwidth utilization without smoothing. Figure 6b highlights the effect of using the MA smoothing process, where *q* = 0.003. Furthermore, Figure 6c presents the effect of applying the MA smoothing technique using *q =* 0.05.

It is evident in Figure 6c that a higher *q* value may cause the loss of the main features of the time series, bearing in mind that in the existing data-centric world losing even a small amount of data can cause a violation of the service-level agreement. It can also lead to inefficient resource planning and utilization. Therefore, *q* must be selected according to Algorithm 1. From Table 5 it is clear that the moving average produced the highest MSE, while LOWESS yielded the second largest MSE due to the likelihood that the first-degree polynomial linear model will not fit the nonlinear bandwidth slice adequately. Fitting using the LOESS-based quadratic polynomial produced a smaller MSE owing to the nonlinearity of the second-order local fitting models. On the other hand, the Savitzky–Golay filter produced a smaller MSE using a second-degree polynomial, compared with the LOESS, where the weights were strongly influenced by *q*, as shown in Equation (6). Lastly, RLOESS and RLOWESS shared a similar performance, yielding the lowest MSE values, as shown in Table 5.

Table 6 shows the performance of combining the local smoothing and LSTM introduced in Algorithm 3. The main objective of this study was to improve the forecast accuracy as a part of better network resource allocation. Therefore, the RMSE was selected as the performance metric. The tables highlight the effects of the proposed methodology on the training and computational time for every bandwidth slice. The improved results were ranked (in brackets) and highlighted accordingly, corresponding to the best combination of algorithms regarding the training RMSE, training time, and testing RMSE for 350 time steps’ forecasting. The results were compared with an earlier study [23] and used as a performance benchmark.

Table 6 and Table 7 showed that the hybrid moving average and LSTM (MLSTM) technique showed the best performance in training and testing the RMSE. However, it may require a higher computation time, such as in the LTE and upstream training phases. These results can be applied to the LTE and MPLS backbone bandwidth traffic, while hybrid RLOWESS and LSTM (RLWLSTM) showed the best performance against the upstream traffic. Although the MLSTM ranking scores were consistently high and showed an average ranking of 1.5, some performance divergence issues can be noted between the different bandwidth slices with other traffic profiles and statistical distributions.

This was further supported by the results presented in Table 3, wherein LTE and MPLS exhibited a similar Johnson SB distribution, while the upstream traffic followed the Gen. extreme distribution. Thus, the data reshaping resulting from the smoothing process can improve the LSTM forecast accuracy, provided that minimum losses can be stripped from the original data. However, no significant improvement in the processing time was noted in some hybrid algorithms, while the process showed some penalties of extra processing time. This drawback can be compensated for by increasing the computation powers of the processing CPU/GPU or routing engines. Figure 7, Figure 8 and Figure 9 show the improvement and degradation in the forecasting accuracy and computational time in terms of percentages compared with the original traffic and results presented in earlier studies [23]. Figure 7a shows the training accuracy improvement for the LTE traffic in terms of percentages. It was noted that MLSTM showed better accuracy (by ≈29%) in the training phase, while the other algorithms showed a lower performance. However, this enhancement required 7% extra computation time. In Figure 7c, the accuracy for the 350 time steps’ forecasting was improved by 22% using MLSTM and the processing time improved by ≈5%, while LLSTM achieved the maximal computational time gain; however, the training and testing performances degraded.

Regarding the MPLS backbone traffic profile, all other hybrid algorithms showed better performance compared with the original profile [18] during the training and testing phase by almost 20% on average, where higher scores were scored by MLSTM and LWLSTM by nearly 35% as depicted in Figure 8. In addition, the computational processing times for MLSTM, LLSTM, and RLLSTM also improved during the training and testing phases, as presented in Figure 8.

Regarding the upstream traffic, a significant improvement was observed for the upstream slices in all the training RMSE values, except RLLSTM. Furthermore, all the algorithms showed better computation times during the training phase (≈8%), except LLSTM. In addition to that, RLWLSTM showed 50% better performance during the testing phase than the other algorithms. Finally, regarding the computational time, only RLWLSTM showed a 4% lower performance, as depicted in Figure 9.

It was seen that the forecasting performance can be improved after using the proposed Algorithms. However, the performance depended on the data series (bandwidth slice) and its statistical properties. Therefore, the dynamic learning framework presented in Figure 5 and Algorithm 3 can help detect any changes occurring in the data distribution. Thus, new hybrid algorithms are to be introduced to replace the old forecasting algorithm. Table 8 compares the forecasting RMSE without the proposed dynamic learning framework presented in Figure 5 and the work presented in Algorithm 3.

Table 8 contains the algorithms obtained from the results in Table 7. It was evident that the proposed framework can detect the changes in the statistical distribution of the slices and provide new hybrid algorithms. The statistical distribution for each slice was obtained from Table 3 (referred to as the actual statistical distribution in Table 8). Compared against different distributions (referred to as new statistical distributions in Table 8), which are already observed within the other slices, the improved performance was 94% for the LTE 350 time steps’ forecast, while for MPLS the improvement percentage was 100%, and finally for the upstream slice the rate was 100%.

## 5. Conclusions

This study used the hybrid local smoothing and LSTM modeling approaches to forecast the bandwidth slice utilization. Six local smoothing techniques were studied: LOWESS, LOESS, moving average, Savitzky–Golay, RLOESS, and RLOWESS. The resultant algorithms, i.e., MLSTM, LLSTM, LWLSTM, SLSTM, RLWLSTM, and RLLSTM indicated that the hybrid LSTM can improve the forecasting accuracy. However, the improvement can be accompanied by the additional computational overhead, and the obtained results may vary depending on the underlying statistical properties of the tested data series. Therefore, the researchers incorporated a dynamic framework to detect and provide a new hybrid algorithm. The results were verified by the statistical significance tests and compared with previous studies. The researchers believed their proposed technique can be used to forecast the 4G/5G and beyond for reliable slice resource management. Furthermore, these results can be extended and applied in the automatic resource allocation algorithm as part of the slice allocator or orchestrator in the 5G networks and beyond.

## Figures and Tables

**Figure 1 sensors-22-03592-f001:**
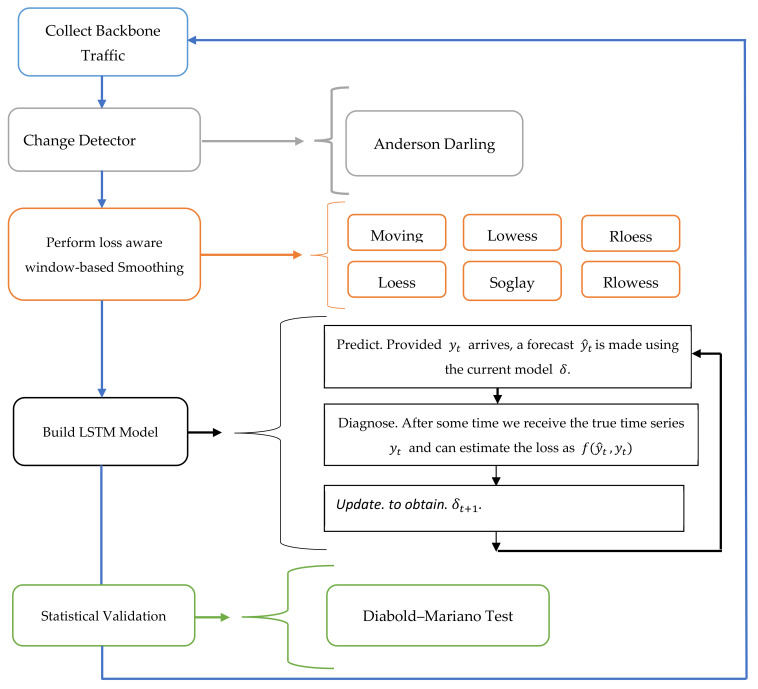
Conceptual framework.

**Figure 2 sensors-22-03592-f002:**
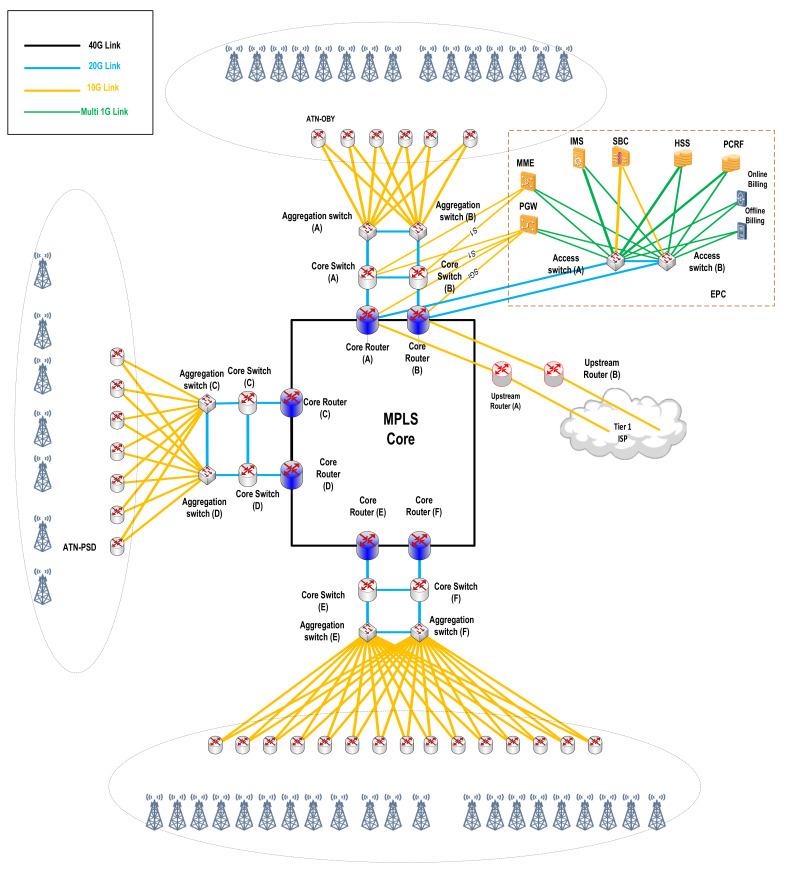
Backbone topology.

**Figure 3 sensors-22-03592-f003:**
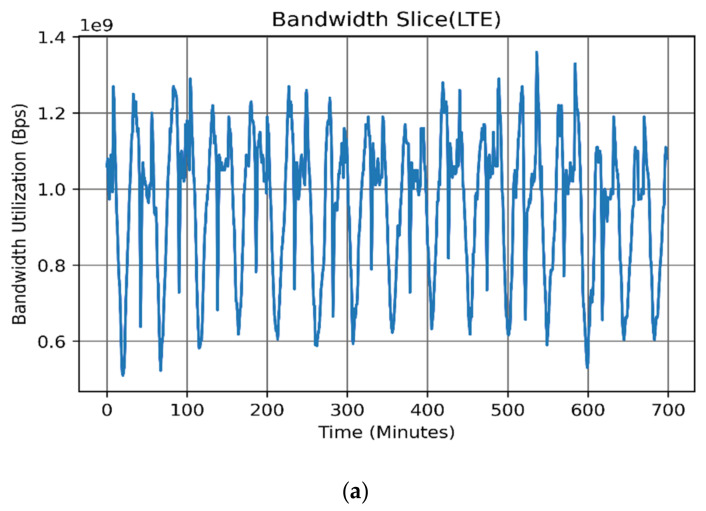
Backbone bandwidth slices: (**a**) LTE, (**b**) MPLS, and (**c**) upstream traffic.

**Figure 4 sensors-22-03592-f004:**
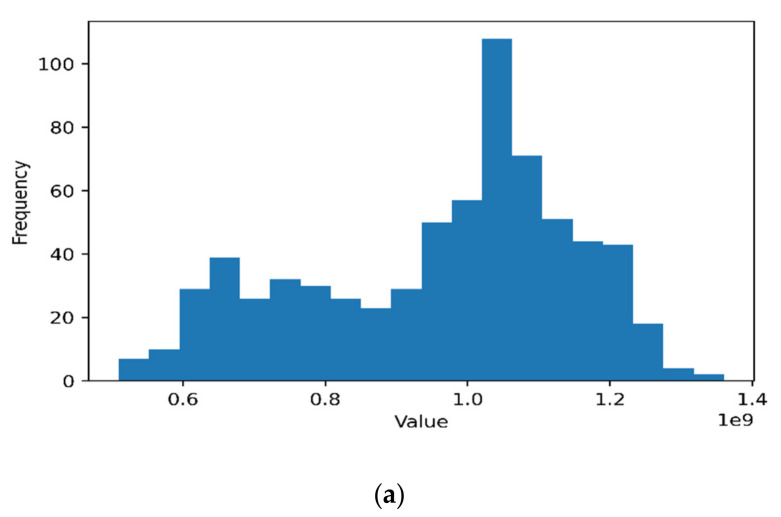
Dataset histograms: (**a**) LTE, (**b**) MPLS, and (**c**) upstream traffic.

**Figure 5 sensors-22-03592-f005:**
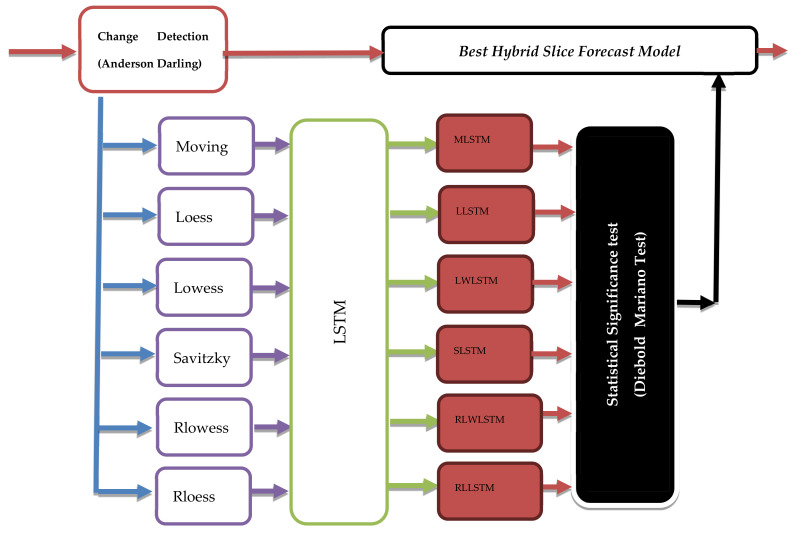
Dynamic learning framework.

**Figure 6 sensors-22-03592-f006:**
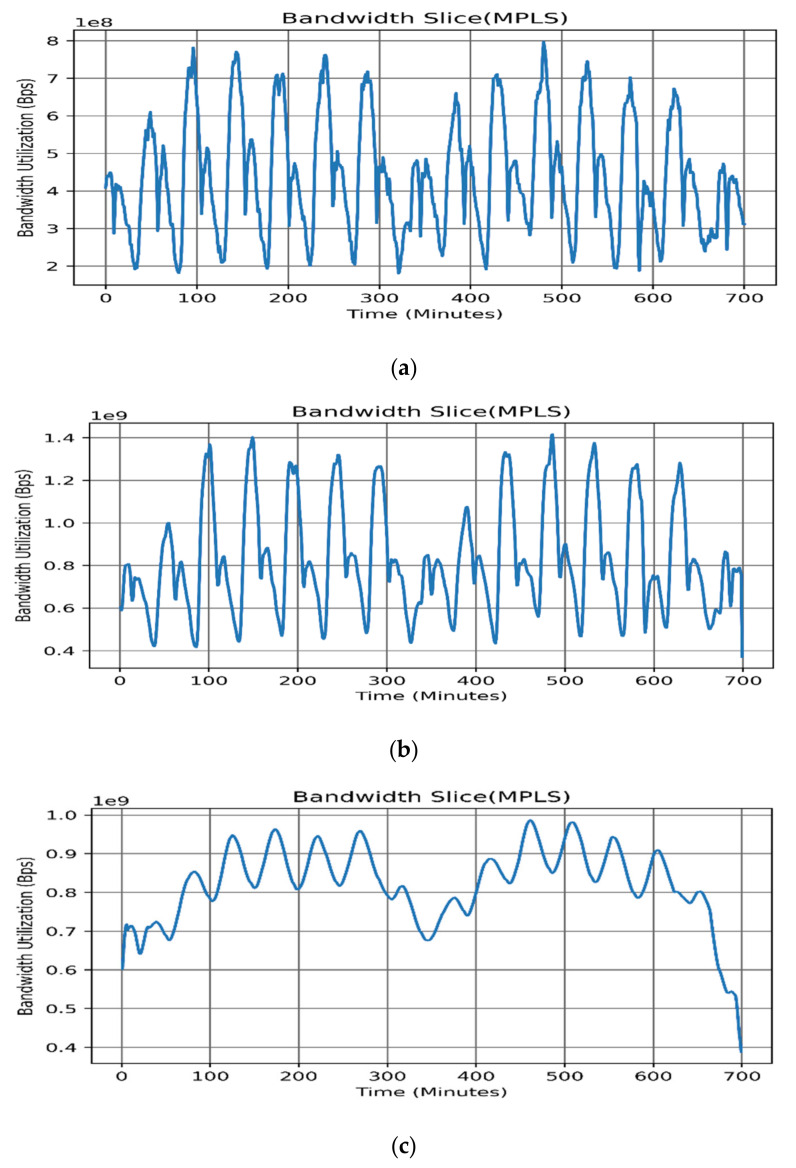
Bandwidth utilization using moving average: (**a**) original MPLS slice (**b**); MPLS slice smoothed with *q =* 0.003; and (**c**) MPLS slice smoothed with *q* = 0.05.

**Figure 7 sensors-22-03592-f007:**
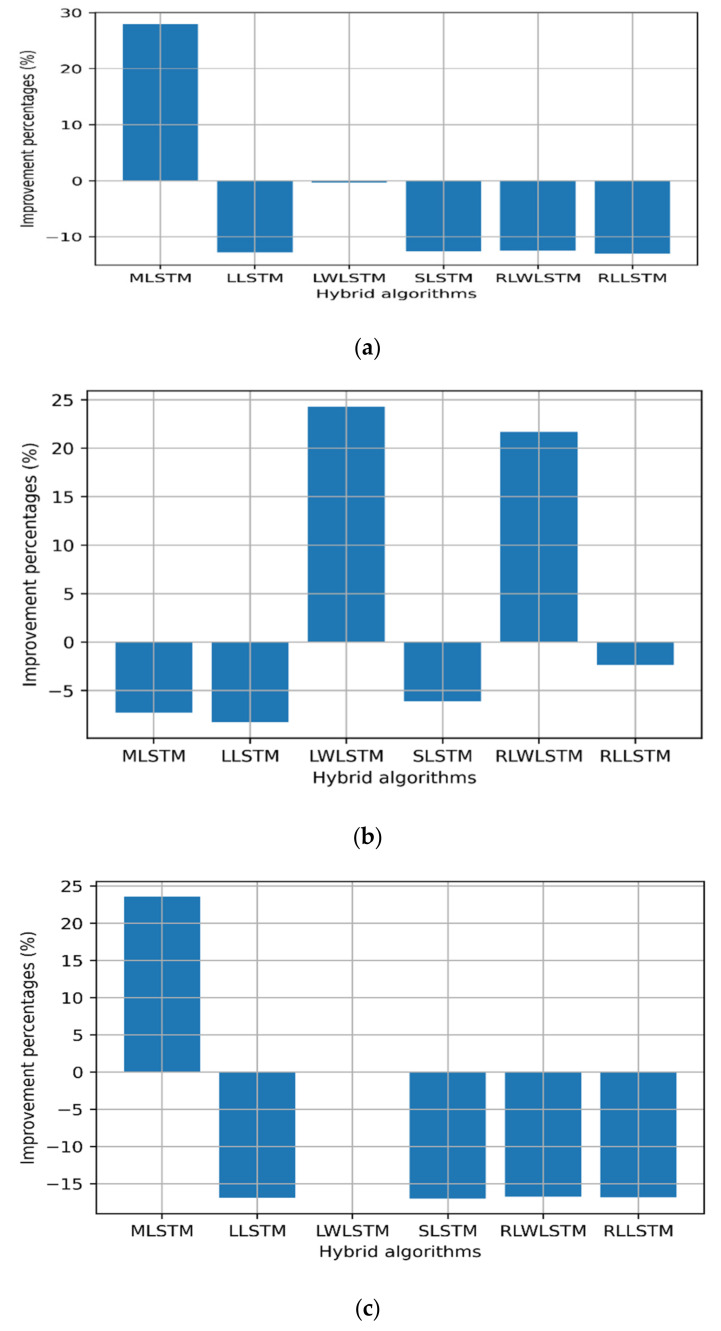
Improvement percentages for LTE: (**a**) training RMSE, (**b**) training time, (**c**) 350 time steps’ testing, and (**d**) 350 time steps’ testing time.

**Figure 8 sensors-22-03592-f008:**
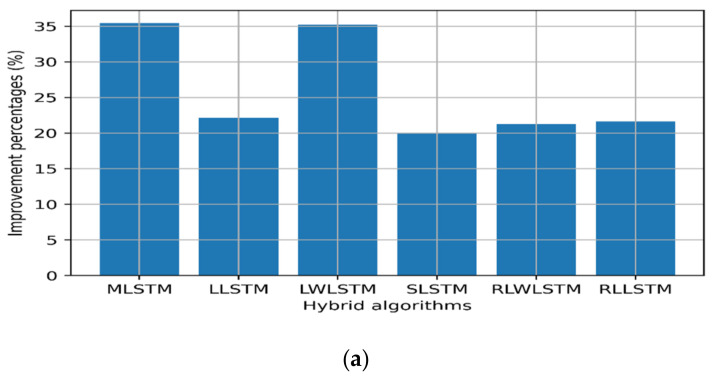
Improvement percentages for MPLS traffic: (**a**) training RMSE, (**b**) training time, (**c**) 350 time steps’ testing, and (**d**) 350 time steps’ testing time.

**Figure 9 sensors-22-03592-f009:**
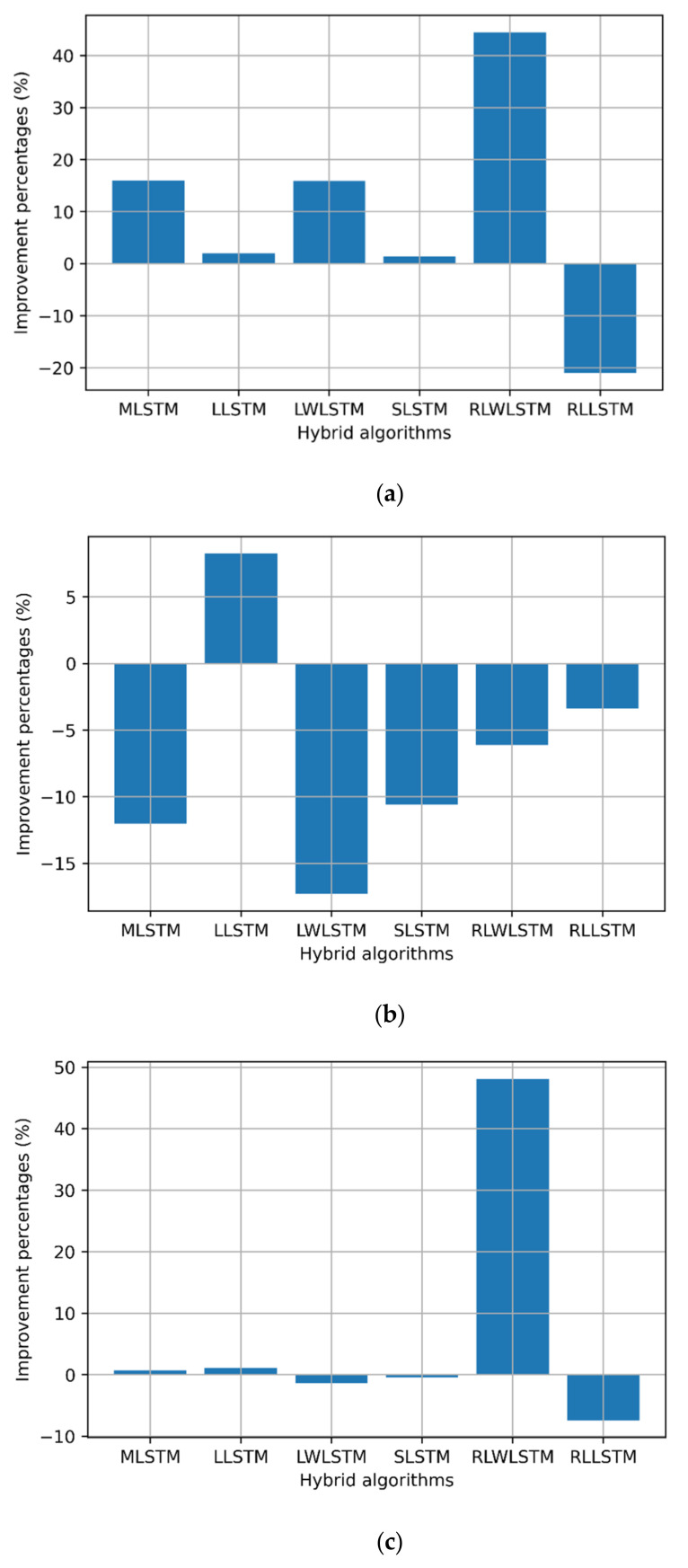
Improvement percentages for upstream: (**a**) training RMSE, (**b**) training time, (**c**) 350 time steps’ testing, and (**d**) 350 time steps’ testing time.

**Table 1 sensors-22-03592-t001:** Related work summary.

Ref	ML Technique	Application (Approach)	Dataset	Noise Preprocessing	Dynamic Learning
[28]	NN, DT, and SVM	Forecast and performance assessment of video over the internet	Internet trace (14-day and 10-day datasets)	No	No
[24]	Back propagation NN	Improvement of network forecasting accuracy	Four days of network traffic	Yes (wavelet)	No
[23]	LSTM	To predict the number of AMFs in 5G core	Control traffic	No	No
[21]	LSTM	To forecast cellular traffic	4G traffic utilization data collected for 122 days	No	No
[22]	LSTM	To forecast (<30 s) tier 1 ISP traffic	Tier 1 ISP traffic variable, hourly, daily, 5 min	Yes (EMD)	No
[29]	LSTM	To forecast network traffic	Network traffic	Yes (EMD)	No
[27,30]	LSTM	To forecast V2V traffic	V2V traffic	No	No
[31]	SVM	To forecast video traffic	Video traffic	Yes (Gaussian smoothing, moving average, and Savitzky–Golay filters)	No
[32]	LSTM and convolutional neural network	To forecast wireless network traffic	Wireless traffic	No	No

**Table 2 sensors-22-03592-t002:** Slice description.

No	Bandwidth Slice	Description
1	LTE	Represents the aggregated backbone bandwidth traffic for 4G-LTE
2	MPLS	Represents the aggregated backbone traffic for corporate data centers
3	Upstream traffic	Represents the aggregated backbone traffic to the tier 1 internet service provider

**Table 3 sensors-22-03592-t003:** Descriptive statistics.

	Sample Size	Range	Mean	Variance	Std. Deviation	Skewness	Min	10%	25% (Q1)	50% (Median)	75% (Q3)	Distribution
LTE	701	9.9 × 10^8^	9.8 × 10^8^	4.5 × 10^16^	2.1 × 10^8^	−0.505	4.58 × 10^8^	6.50 × 10^8^	8.2 × 10^8^	1.0 × 10^9^	1.13 × 10^9^	Johnson SB
MPLS	701	6.1 × 10^8^	4.4 × 10^8^	2.3 × 10^16^	1.5 × 10^8^	-	1.80 × 10^8^	2.43 × 10^8^	3.1 × 10^8^	4.3 × 10^8^	5.48 × 10^8^	Johnson SB
Upstream	701	4.75 × 10^9^	5.1 × 10^9^	9.76 × 10^17^	9.8 × 10^8^	−0.463	2.67 × 10^9^	3.57 × 10^9^	4.4 × 10^9^	5.3 × 10^8^	5.79 × 10^9^	Gen. extreme value

**Table 4 sensors-22-03592-t004:** LSTM hyperparameters.

Parameter	Name
Library (Python)	Tensorflow, Keras, NumPy, Sklearn
Batch size	1
Epochs	20
Optimizer/learning rate	ADAM
Loss function	RMSE
Neurons	2
Hidden layer	1
Activation function	ReLU

**Table 5 sensors-22-03592-t005:** Smoothing MSE.

Smoothing Technique	LTE-MSE	MPLS-MSE	Upstream MSE
Moving average	2.41 × 10^7^	4.77 × 10^7^	7.89 × 10^7^
LOWESS	2.0785 × 10^7^	2.65 × 10^7^	6.79 × 10^7^
LOESS	6.40 × 10^4^	1.40 × 10^7^	1.10 × 10^5^
SGolay	1.0133 × 10^−8^	2.17 × 10^−9^	2.25 × 10^−8^
RLOWESS	1.7030 × 10^−10^	1.70 × 10^−10^	1.03 × 10^8^
RLOESS	1.7030 × 10^−10^	1.70 × 10^−10^	7.03 × 10^7^

**Table 6 sensors-22-03592-t006:** Performance of combining local smoothing and LSTM.

Slice	Smoothing Technique	Training RMSE	Training Time (s)	Testing RMSE for 350 Time Steps	Testing Time for 350 Time Steps		Smoothing Technique	Training RMSE	Training Time	Testing RMSE for 350 Time Steps	Testing Time for 350 Time Steps
LTE	Original [23]	8062987	12.933	7395539	0.00560	MPLS	Original [23]	5400982	7.365	4982949	0.0059
	MLSTM	5783686 (1)	13.872	56527763 (1)	0.00531 (3)		MLSTM	3486200 (1)	7.221 (3)	3380349 (1)	0.0049 (3)
	LLSTM	9092992	14.004	8644144	0.0045 (1)		LLSTM	4205124 (3)	6.849 (1)	4022174 (3)	0.0039 (1)
	LWLSTM	8065674	9.785 (1)	7391435 (2)	0.0051 (2)		LWLSTM	3496462 (2)	8.5821	3411503(2)	0.0059 (5)
	SLSTM	9075783	13.725	8655062	0.0061		SLSTM	4286874 (6)	7.588	4071104 (6)	0.0049 (3)
	RLWLSTM	9067949	10.120 (2)	8635038	0.0054 (5)		RLWLSTM	4250014 (5)	6.979 (2)	4045647 (4)	0.0040 (2)
	RLLSTM	9112072	13.237	8640586	0.00535 (4)		RLLSTM	4232122 (4)	13.995	4045664 (5)	0.0058 (4)

**Table 7 sensors-22-03592-t007:** Performance of combining local smoothing and LSTM.

Slice	Smoothing Technique	Training RMSE	Training Time(s)	Testing RMSE for 350 Time Steps	Testing Time (s) 350 Time Steps
Upstream	Original [23]	4056533	12.367	3836587	0.005761
	MLSTM	3403134 (2)	13.896	3810110 (3)	0.00530 (3)
	LLSTM	3976124 (4)	11.346 (1)	3792150 (2)	0.00524 (2)
	LWLSTM	3411313 (3)	14.517	3889609 (5)	0.00576 (5)
	SLSTM	3997700 (5)	13.687	3819971 (4)	0.00521 (1)
	RLWLSTM	2253007 (1)	13.126	1989996 (1)	0.006011
	RLLSTM	4946024	12.786	4122146	0.00544 (4)

**Table 8 sensors-22-03592-t008:** Performance of dynamic learning.

Slice	Techniques	Actual Statistical Distribution	New Statistical Distribution	Without Dynamic Learning Framework (RMSE)	With Dynamic Learning Framework (RMSE)
LTE	MLSTM	Johnson SB	Gen. gamma (4P)	962141749	56527763 (94%)
MPLS	MLSTM	Johnson SB	Gen. extreme value	4752069825	3380349 (100%)
Upstream	RLWLSTM	Gen. extreme value	Gen. gamma (4P)	5157991293	1989996 (100%)

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
