# Peer review of "Dynamic Learning Framework for Smooth-Aided Machine-Learning-Based Backbone Traffic Forecasts"

_sensors, 2022, doi:10.3390/s22093592_

Round 1
Reviewer 1 Report
Resource allocation and QoS are certainly the core issues in networking resource planning. The authors proposed a hybrid algorithm, which combines change detection, smoothing algorithms and LSTM, for network traffic forecasting. The algorithm is worth investigating, but the listed algorithms (especially the Algorithm 3), results, and discussion need to be explained clearly. Some major and minor issues are listed below.
Main issues:
- The workflow presented in Figure1 doesn’t match the workflow shown in Figure 4.
- Would the Algorithm 3 be much efficient if ? is cached rather than replaced?
- Page 13, the Algorithm 2 does not provide any valuable information to the proposed framework.
- In Table 3, can the authors confirm that 1 hidden layer and especially the 2 neurons were used in the LSTM?
- For Algorithm 3 used in the Results and Discussion, how many data points are in each “time step” and for each “Wj”? Where was the initial model (v) built (Line 09)?
- Page 13, Line 482-484, the author stated that “Here, change refers to a change in statistical distribution between Wj and Wj+1,…”, then in Algorithm 3, (1) “Stop forecasting at Wj” (Line 06), (2) “y hat <- smoothy in Wj+1…” (Line 07); (3)”E <- calculate Forecast error of ti in Wj+2..” (Line 10), (4) “replace ? i-1 by ?”. Following the algorithm, if the distribution of Wj continuously changes, when will the forecast be performed?
- Figure 6, 7, 8 show repeated information as given in Table.
- Page 19, Line 593, where does the accuracy, “MLSTM showed a better accuracy (by approx. 29%”, come from?
- In Table 5, what is the Training RMSE? Do the authors have the Testing RMSE?
- In Table 6 and the description given in Line 689 – 697, it is hard to think how these attributes, i.e., the statistical distribution, new statistical distribution, without dynamic learning framework (RMSE), with dynamic learning framework (RMSE), could be interrupted and related.
Minor Issues:
- Page 2, Line 54, change “intracTable” to ‘intractable’
- Page 3, Line 104-105, explain what the term “an employee change detection process”
- Page 5, Line 241, change “coolected” to ‘collected’
- In Algorithm 3, what does Line 04: “Wj <- Wj U {ti}” mean in the pseudocode? The same question for Line 11
- The alignment of lines in Algorithm 3 should be formatted properly
- Page 14, Line 489, change “By utilizing”
Author Response
April 27, 2022
Original Manuscript ID: sensors-1631570
Original Article Title:(Dynamic Learning Framework for Smooth-Aided Machine-Learning-Based Backbone Traffic Forecasts)
To: MDPI sensor/Sensors/ Reviewer 1
Re: Response to Reviewer 1
We thank you for the opportunity to revise our manuscript. We appreciate the careful review and constructive suggestions. It is our belief that the manuscript is substantially improved after making the suggested edits. Following this letter, we are uploading our point-by-point responses to your comments. In addition to that, the manuscript is uploaded in a separate link where changes were marked using track changes. The manuscript language and layout have been reviewed by MDPI English and layout service
Sincerely,
Mohamed Khalafalla Hassan
(Corresponding author)

Reviewer 2 Report
Review:
In this study, the authors proposed a reliable hybrid dynamic network-bandwidth slice forecasting framework which is based on LSTM neural network and Local smoothing methods. For improving the network forecasting, they utilized local smoothing methods, and it has Lowess, Loess, Moving average, Savitzky Golay, Rloess, and Rlowess. Backbone traffic is used for validating the proposed work.
Strengths:
The results have shown that MLSTM achieved the most remarkable improvement in training and testing forecast with 28% and 24% for LTE time series, and with 35% and 32% for MPLS time series, respectively, while RLWLSTM achieved the most significant improvement for Up-stream traffic with 45% for training and testing RMSE.
Weakness:
The manuscript requires revision based on identified issues present in the proposed framework.
1. In the Abstract, provide the full abbreviation of MLSTM, RLWLSTM, RMSE, LTE, MPLS, and revised the Abstract. Improve English also, because the Abstract should be crunchy for any manuscript.
2. Line 29 and 30, use the capital letter Rlowess, Rloess, and others.
3. Line 34 to 38, are divided into two or three simple sentences.
4. Line 40, check journal format.
5. According to the title of the manuscript, motivation is not sufficient in Section 1.
6. Add a comparison table in Section 2, and show the limitations of the existing work and how to resolve it with the proposed work. Add a Summary paragraph also add after the comparison table.
7. Some symbols are used in Fig. 1. What is this? It is tough for understanding the work. Revise it carefully. What is Anderson Darling, add a description in Fig. 1.
8. Dataset description provides in the tabular form in Section 3.
9. What are EPC and MPLS in Fig. 2? Add description.
10. Improve the visible quality of Fig. 3 and 5.
11. Why are you using MPLS backbone network? Explain it in Subsection 3.1.
12. What is MSE in Algorithm 1? Provide proper format of all algorithms such as What is Input, Output, and Process?
13. Possibly, provide an abbreviation table before Section 3 for a better understanding of the manuscript.
14. Check Algorithm 2 & 3, formatting issues.
15. Provide more details in the evaluation section.
16. The authors should remove the grammatical mistakes and typos in the paper.
17. Cross-reference all citations and ensure that they match accordingly. The reference paper format should be uniform. I have identified some of the references with missing details like page numbers, volume numbers, issue numbers, etc. Recent reference must be added, the following is recommended:
• KST-GCN: A Knowledge-Driven Spatial-Temporal Graph Convolutional Network for Traffic Forecasting.
• Machine learning-based network sub-slicing framework in a sustainable 5g environment.
• Forecasting Network Interface Flow Using a Broad Learning System Based on the Sparrow Search Algorithm
• Machine learning based distributed big data analysis framework for next generation web in IoT.
• Prediction of Network Traffic in Wireless Mesh Networks using Hybrid Deep Learning Model.
Author Response
April 27, 2022
Original Manuscript ID: sensors-1631570
Original Article Title:(Dynamic Learning Framework for Smooth-Aided Machine-Learning-Based Backbone Traffic Forecasts)
To: MDPI sensor/Sensors/ Reviewer 2
Re: Response to Reviewer 2
We thank you for the opportunity to revise our manuscript. We appreciate the careful review and constructive suggestions. It is our belief that the manuscript is substantially improved after making the suggested edits. Following this letter, we are uploading our point-by-point responses to your comments. In addition to that, the manuscript is uploaded in separate link where changes were marked using track changes. The manuscript language and layout have been reviewed by MDPI English and layout service
Sincerely,
Mohamed Khalafalla Hassan
(Corresponding author)

Reviewer 3 Report
The article is focused on dynamic network backbone traffic forecasting by means of LSTM neural networks and local smoothing techniques.
In section 2 a detailed analysis of the related work is given, quite fresh references are regarded.
In page 3 a detailed description of the contribution is presented. However it lacks due comments why this method is really hybrid. Besides it lacks a proof of the novelty in comparison with these related works considered.
Figure 1 lacks arrow semantics. In addition, there is a lack of clarifications for some of the mathematical terms used in Figure 1.
In Figure 3 there are missing units on the horizontal axis. They should be added.
In addition, section 3.1 lacks characteristic fragments of the initial data for the three types of slices under consideration. It is also not justified how typical and independent the selected dataset is. Are the time patterns (presented in Figure 3) the result of a very particular case study? Hypothetically the time patterns presented in Figure 3 could be a result of a very particular case study. One should check it more deeply.
In Section 3.1 it is stated that got empirical data meet Johnson SB statistical distribution (for LTE and MPLS) and Gen. Extreme Value distribution (for Upstream traffic). However, it is not clear how close the available data actually are to these distributions. And how did you decide to check the data for belonging to exactly these particular distributions? Possibly you enumerated some list of potential ones? Or you expected to try anything specific based on your previous experience and/or some work in the field?
Table 3 exposes the resulted LSTM hyper-parameters selected by GridSearch. However it lacks a description of possible options (or bounds) considered during this selection process.
In the experimental part, the main attention is paid to the performance-related metrics of the developed combined methods. At the same time there is no clear diagram showing the values of forecasting quality metrics.
The article should be subjected to minor format revision. In page 13 the numbers of the lines ran into the Algorithm 2 and Algorithm 3, it is hard to read.
It seems the format of the article should be checked, some of the illustrations are printed outside the margins. It should be checked for misprints as well (e.g. in line 34).
Author Response
April 27, 2022
Original Manuscript ID: sensors-1631570
Original Article Title:(Dynamic Learning Framework for Smooth-Aided Machine-Learning-Based Backbone Traffic Forecasts)
To: MDPI sensor/Sensors/ Reviewer 3
Re: Response to Reviewer 3
We thank you for the opportunity to revise our manuscript. We appreciate the careful review and constructive suggestions. It is our belief that the manuscript is substantially improved after making the suggested edits. Following this letter, we are uploading our point-by-point responses to your comments. In addition to that, the manuscript is uploaded in separate link where changes were marked using track changes. The manuscript language and layout have been reviewed by MDPI English and layout service
Sincerely,
Mohamed Khalafalla Hassan
(Corresponding author)

Reviewer 4 Report
The authors must conduct exploratory data analysis, such as histograms, scatter plots, and box plots, or simple explanations for the reader to understand, such as concept links and cluster analysis tables.
The description of the input variable is unclear. Using a table, the authors should clearly explain the input and output variables. What input variables did the authors consider?
- The authors must exhibit correlation metrics (p-value) for input, exogenous, and output variables. In addition, the authors should tabulate the input variables the authors used in this paper along with their data types and units.
Did the authors consider the Min-Max normalization in Section 3?
I wonder if the proposed method can be applied to other regions with different mechanical systems and occupant profiles. It is challenging to trust the results of this experiment fully. In addition, deep learning analysis requires repeated experiments because the predicted value varies depending on how the initial weight is set.
This problem is complex to tackle, given the limited memory and storage of mobile devices and why only ML is fit for the task. This element requires extra attention in the Discussion section of the paper, referring to the following:
- Md Ibrahim Khalil, R. Young Chul Kim, ChaeYun Seo (2020). Challenges and Opportunities of Big Data. JOURNAL OF PLATFORM TECHNOLOGY, 8(2), 3-9;
- S.Vimal, Y. Harold Robinson, M.Kaliappan, Subbulakshmi Pasupathi, A.Suresh (2021). Q Learning MDP Approach to Mitigate Jamming Attack Using Stochastic Game Theory Modelling With WQLA in Cognitive Radio Networks. JOURNAL OF PLATFORM TECHNOLOGY, 9(1), 3-14;
- S.Vimal, Jesuva Arockiadoss S, Bharathiraja S, Guru S, V.Jackins (2021). REDUCING LATENCY IN SMART MANUFACTURING SERVICE SYSTEM USING EDGE COMPUTING. JOURNAL OF PLATFORM TECHNOLOGY, 9(1), 15-22;
- Han, Y., & Hong, B. W. (2021). Deep learning based on Fourier convolutional neural network incorporating random kernels. Electronics, 10(16), 2004.
- Kumar, R. L., Khan, F., Kadry, S., & Rho, S. (2022). A Survey on blockchain for industrial Internet of Things. Alexandria Engineering Journal, 61(8), 6001-6022.
The authors can consult the following references for statistical tests, such as the Wilcoxon signed-rank and Freidman tests.
- Rew, J., Cho, Y., Moon, J., & Hwang, E. (2020). Habitat suitability estimation using a two-stage ensemble approach. Remote Sensing, 12(9), 1475.
The authors must improve all figures in quality.
According to the journal format, the authors should present words such as abbreviations in advance.
Author Response
April 27, 2022
Original Manuscript ID: sensors-1631570
Original Article Title:(Dynamic Learning Framework for Smooth-Aided Machine-Learning-Based Backbone Traffic Forecasts)
To: MDPI sensor/Sensors/ Reviewer 4
Re: Response to Reviewer 4
We thank you for the opportunity to revise our manuscript. We appreciate the careful review and constructive suggestions. It is our belief that the manuscript is substantially improved after making the suggested edits. Following this letter, we are uploading our point-by-point responses to your comments. In addition to that, the manuscript is uploaded in a separate link where changes were marked using track changes. The manuscript language and layout have been reviewed by MDPI English and layout service
Sincerely,
Mohamed Khalafalla Hassan
(Corresponding author)

Round 2
Reviewer 1 Report
- Figure 1 is missing
- The explanation to Item 6 (given in the previous comment) should further be improved.
Author Response
April 29, 2022
Original Manuscript ID: sensors-1631570
Original Article Title:(Dynamic Learning Framework for Smooth-Aided Machine-Learning-Based Backbone Traffic Forecasts)
To: MDPI sensor/Sensors/ Reviewer 1
Re: Response to Reviewer 1 (Round 2)
We thank you for the opportunity to revise our manuscript. We appreciate the careful review and constructive suggestions. It is our belief that the manuscript is substantially improved after making the suggested edits. Following this letter, we are uploading our point-by-point responses to your comments. In addition to that, the manuscript is uploaded in separate link where changes were marked using track changes. The manuscript language and layout have been reviewed by MDPI English and layout service
Sincerely,
Mohamed Khalafalla Hassan
(Coresponding author)
Response to Reviewer 1 Comments
Point 1: Figure 1 is missing
Response 1: Figure 1 has been added.
Point 2: The explanation of Item 6 (in the previous comment) should be further improved.
Response 2: Point 6 was
Page 13, Line 482-484, the author stated that “Here, change refers to a change in statistical distribution between Wj and Wj+1,…”, then in Algorithm 3, (1) “Stop forecasting at Wj” (Line 06), (2) “y hat <- smoothy in Wj+1…” (Line 07); (3)” E <- calculate Forecast error of ti in Wj+2..” (Line 10), (4) “replace ? i-1 by ?”. Following the algorithm, if the distribution of Wj continuously changes, when will the forecast be performed?
Response 6: If the data distribution keeps changing, the system will stop using the old hybrid algorithm Furthermore, no more new forecasts will be produced. Until at least the distribution in two successive windows and is the same. Because if we trained the ML model on a sample set, then the data distribution is changed. The ML will produce less accurate and less reliable forecasted data. This is the main reason for deploying the change detector. This issue is one of the central liabilities of applying ML in real life. Data streams can change, and ML algorithms alone can not capture these changes and adapt themselves dynamically and continuously . In the Case of the backbone traffic (our Case), it is unlikely that the distribution will be continuously changing during the selected window although it is possible, due to nature of traffic profiles that aggregates large number of services/ users

Reviewer 3 Report
A related work summary and its analysis have been added to section 2. Specific factors, the peculiarities of the solutions were highlighted and extra comments proving the novelty have been added. Figure 1 has been improved, an extra data on it added. Missing labels of units along horizontal axis have been added into Figure 3, now it becomes clearer. As for Section 3.1 it lacks an example of data from the dataset, please show its a few the most typical samples. For more completeness please expose the name of the software you used to find the best statistical distribution fit for the data series and used functions. Extra statistics have been presented in Table 3. The format of the article has been corrected, no that graphical overlapping is no longer presented in illustrations.
Author Response
April 29, 2022
Original Manuscript ID: sensors-1631570
Original Article Title:(Dynamic Learning Framework for Smooth-Aided Machine-Learning-Based Backbone Traffic Forecasts)
To: MDPI sensor/Sensors/ Reviewer 3
Re: Response to Reviewer 3 (Round 2)
We thank you for the opportunity to revise our manuscript. We appreciate the careful review and constructive suggestions. It is our belief that the manuscript is substantially improved after making the suggested edits. Following this letter, we are uploading our point-by-point responses to your comments. In addition to that, the manuscript is uploaded in separate link where changes were marked using track changes. The manuscript language and layout have been reviewed by MDPI English and layout service
Sincerely,
Mohamed Khalafalla Hassan
(Coresponding author)
Response to Reviewer 3 Comments
Point 1: A related work summary and its analysis have been added to section 2. Specific factors, the peculiarities of the solutions were highlighted and extra comments proving the novelty have been added. Figure 1 has been improved, an extra data on it added. Missing labels of units along horizontal axis have been added into Figure 3, now it becomes clearer. As for Section 3.1 it lacks an example of data from the dataset, please show its a few the most typical samples. For more completeness please expose the name of the software you used to find the best statistical distribution fit for the data series and used functions. Extra statistics have been presented in Table 3. The format of the article has been corrected, no that graphical overlapping is no longer presented in illustrations
Response 1: Thanks for your comments. The Dataset examples have been shown in figure 3 (a)(b)(c). Regarding the second part, the name of the software is EasyFit
